# Beyond Reward Hacking: Causal Rewards for Large Language Model Alignment

## Abstract

Recent advances in large language models (LLMs) have demonstrated significant progress in performing complex tasks. While Reinforcement Learning from Human Feedback (RLHF) has been effective in aligning LLMs with human preferences, it is susceptible to spurious correlations in reward modeling. Consequently, it often introduces biases—such as length bias, sycophancy, conceptual bias, and discrimination—that hinder the model's ability to capture true causal relationships. To address this, we propose a novel causal reward modeling approach that integrates causality to mitigate these spurious correlations. Our method enforces counterfactual invariance, ensuring reward predictions remain consistent when irrelevant variables are altered. Through experiments on both synthetic and real-world datasets, we show that our approach mitigates various types of spurious correlations effectively, resulting in more reliable and fair alignment of LLMs with human preferences. As a drop-in enhancement to the existing RLHF workflow, our causal reward modeling provides a practical way to improve the trustworthiness and fairness of LLM finetuning.

## 1 Introduction

Recent advancements in large language models (LLMs) have demonstrated remarkable capabilities in generating coherent, contextually appropriate responses across a wide range of tasks (Brown et al., 2020). A key approach to further refine these models is Reinforcement Learning from Human Feedback (RLHF), which leverages human evaluations to guide the training process and align model outputs more closely with human preferences (Stiennon et al., 2020; Ouyang et al., 2022; Bai et al., 2022; Wang et al., 2024). RLHF typically involves training a reward model to capture human preferences, which is then used to fine-tune LLMs via reinforcement learning (RL) (Schulman et al., 2017; Chen et al., 2024b;f).

Despite the success of RLHF, reward modeling is inherently prone to *spurious correlations*, which are associations in the training data that do not reflect true causal relationships (Veitch et al., 2021), and can lead to unintended biases and induce *reward hacking* (McMilin, 2022). Reward hacking occurs when RL agents exploit flaws or ambiguities in the reward function to maximize rewards without genuinely improving alignment with desired behaviors or completing designed tasks (Amodei et al., 2016; Weng, 2024). Consequently, this leads to misaligned models that exhibit biases such as favoring longer outputs (*length bias*) (Zheng et al., 2023), agreeing with user's incorrect assertions (*sycophancy bias*) (Perez et al., 2022), developing unintended shortcuts when making predictions (*concept bias*) (Zhou et al., 2023), and implicitly developing discrimination over certain demographic groups (*discrimination bias*) (Tamkin et al., 2023; Chen et al., 2024c). These biases, rooted in spurious correlations and reward hacking rather than true causal relationships, undermine the reliability and trustworthiness of LLMs, posing significant challenges for their safe and responsible deployment in real-world applications (Anwar et al., 2024; Qi et al., 2024).

To understand and mitigate these issues, it is essential to consider the sources of error in reward modeling. The total error in the reward model can be decomposed into *reducible* and *irreducible* components. The reducible error comprises estimation errors stemming from limited data and model approximation, which can be alleviated by collecting more data or increasing model capacity. However, irreducible error originates from inherent noise and imperfections in the data, such as the spurious correlations described earlier, which cannot be resolved merely by increasing data quantity or model complexity (Geman et al., 1992). For instance, if longer responses are disproportionately

represented and favored among higher-reward examples, the reward model may learn to prefer longer outputs irrespective of their quality, leading to the length bias observed in RLHF policies. Similarly, human annotators may unintentionally favor responses that flatter them. This bias can mislead the model, causing it to prefer agreeableness over truthfulness (Perez et al., 2022). Notably, such biases cannot be mitigated by simply increasing the size of the dataset. On the contrary, it may further exacerbate the effects of reward hacking (Ribeiro et al., 2016).

To address this challenge, we propose a novel approach in this work that integrates causality into reward modeling to mitigate the impact of spurious correlations and prevent reward hacking in RLHF. By leveraging techniques from causality, we develop a **causal reward model (`CRM`)** that is robust to these spurious correlations and captures the true causal relationship of responses on human preferences. Central to our method is the concept of counterfactual invariance, which ensures that the reward model's predictions remain consistent under interventions on irrelevant aspects of the input, thereby reducing the irreducible error caused by spurious correlations (Veitch et al., 2021).

By addressing the irreducible errors due to spurious correlations, our approach mitigates reward hacking and advances the development of more aligned and trustworthy LLMs, enabling broader adoption in applications that demands high reliability and fairness. Specifically, our contributions can be summarized as follows:

- We introduce a causal framework for reward modeling that incorporates causal regularization into the training process, allowing the model to learn *true*[1] causality from spurious relationship.
- Through experiments on both synthetic and real-world datasets, we demonstrate the effectiveness of our causal reward model (CRM) in mitigating biases, including length, sycophancy, concept, and discrimination biases, which are common factors that lead to reward hacking.
- `CRM` is simple to implement and can be seamlessly integrated into existing RLHF pipelines, providing a practical solution to enhance the reliability of LLMs.

## 2 RELATED WORKS

### 2.1 REWARD HACKING AND SPURIOUS CORRELATION

The issue of reward hacking has become increasingly significant as RLHF grows in popularity over the recent years (Amodei et al., 2016; Casper et al., 2023; Kaufmann et al., 2023; OpenAI, 2023). RLHF aligns LLMs with human preferences by training a reward model (RM) to provide feedback based on user prompts (Christiano et al., 2017; Ziegler et al., 2019; Chen et al., 2024b; Zhang et al., 2024b). However, RMs are often imperfect proxies of the underlying true human preferences, leading to instances of reward over-optimization (Coste et al., 2023; Moskovitz et al., 2023), or *reward hacking* (Denison et al., 2024; Everitt et al., 2021), where models achieve high rewards without fulfilling the intended objectives (Pan et al., 2022; Weng, 2024).

LLM reward hacking often stems from the model's reliance on *spurious correlations* in the preference dataset, such as length (Sountsov & Sarawagi, 2016; Dubois et al., 2024a; Huang et al., 2024), sycophancy (Sharma et al., 2023; Ranaldi & Pucci, 2023), conceptual (Zhou et al., 2023), and demographic (Salinas et al., 2023) biases. These spurious correlations, closely linked to reward hacking, can impair a model's capability to learn and generalize to broader scenarios (Ribeiro et al., 2016; Geirhos et al., 2020; Chen et al., 2024d). Without proper constraints, models will exploit all available informative features during training, including unreliable spurious ones, which results in reward hacking, even if the task is very simple (Nagarajan et al., 2020; McMilin, 2022; Chen et al., 2024e). To address this, our approach integrates causal regularization into reward modeling, enabling LLMs to learn true causal relationships, mitigate the effects from spurious correlations and thereby prevent reward hacking.

### 2.2 ALLEVIATING SPURIOUS CORRELATIONS

Early efforts to mitigate spurious correlations and reward hacking in RLHF have primarily focused on penalizing specific biases within reward models (Mnih et al., 2015), especially correcting for length bias. For example, Singhal et al. (2023) reveals that length-based biases in reward models significantly influence RLHF outcomes, often overshadowing non-length-related features, and proposes mitigation strategies such as balanced preference datasets, reward data augmentation, confidence-based truncation, increased KL penalties, explicit length penalties, omitting long outputs, and focusing

---

[1]Here, *true* represents the user's belief about what is true.

on non-length reward metrics. To address the overemphasis on longer response, Shen et al. (2023) proposed a Product-of-Experts (PoE) framework to decouple reward modeling from sequence length, thereby reducing the reward model's preference for verbose but low-quality responses. Building on this, Eisenstein et al. (2023) introduced reward model ensembles to moderate reward hacking by diversifying the sources of feedback and reducing reliance on any single reward model's spurious correlations. However, this method only partially mitigates the problem and falls short of fully eliminating reward hacking. More recently, Ramé et al. (2024) proposed Weight Averaged Reward Models (WARM), which enhance robustness to distribution shifts by averaging model weights, offering a more efficient and effective alternative to ensemble-based policy interpolation. ODIN (Chen et al., 2024a) advanced this line of work by introducing a disentangled reward model architecture to tackle length bias. Their approach separates reward factors into two linear heads, isolating content quality for use during RL fine-tuning, thus improving performance without sacrificing efficiency.

In contrast to these approaches, our causal reward modeling incorporates causal regularization directly into the reward modeling process. By enforcing counterfactual invariance, we ensure that model responses align with the true causal effects of human preferences rather than being driven by spurious correlations. Notably, unlike existing methods (Singhal et al., 2023; Chen et al., 2024a) that address a single type of spurious correlation, our approach fundamentally mitigates a broad spectrum of spurious correlations, providing a comprehensive solution to reward hacking and enabling more reliable alignment with human preferences.

## 3 PRELIMINARIES

### 3.1 REINFORCEMENT LEARNING FROM HUMAN FEEDBACKS (RLHF)

**Supervised fine-tuning (SFT).** The SFT step typically starts with a pre-trained language model, which is then fine-tuned through supervised learning on a high-quality dataset tailored to specific downstream tasks, such as dialogue (Bai et al., 2022), instruction following (Longpre et al., 2023), and summarization (Zheng et al., 2024). This fine-tuning process produces a model denoted as $\pi^{\text{SFT}}$.

**Reward model learning.** During this stage, we will first need to have a dataset that consists of preference pairs of responses, $(y_1, y_2)$, for each prompt $x$. Typically, these pairs are obtained by presenting them to labelers (e.g., humans), who evaluate the responses based on their preferences, represented as $y_w \succ y_l \mid x$, where $y_w$ and $y_l$ denote the preferred and less preferred responses, respectively. From a modeling perspective, these preferences are assumed to be generated from an unknown latent reward model, $r^*(y, x)$. In practice, the modeling assumptions for the preferences can vary depending on the problem, but the Bradley-Terry (BT) model is a commonly used assumption. The BT model computes the probability of one response $y_1$ being preferred over the other response $y_2$ under the true reward function $r^\star(x, y)$ by:

$$p^*(y_1 \succ y_2|x) = \frac{\exp(r^*(x, y_1))}{\exp(r^*(x, y_1)) + \exp(r^*(x, y_2))}$$

Given a static dataset of preference data $\mathcal{D} = \{x^{(i)}, y_w^{(i)}, y_l^{(i)}\}_{i=1}^N$ sampled from $p^*$, we can fit a reward model $r_\phi(x, y)$ to estimate its parameters through maximum likelihood estimation. This approach is equivalent to binary classification and can be trained by minimizing the negative log-likelihood loss:

$$\mathcal{L}_R(r_\phi, \mathcal{D}) = -\mathbb{E}_{(x, y_w, y_l) \sim \mathcal{D}} \left[ \log \sigma \big( r_\phi(x, y_w) - r_\phi(x, y_l) \big) \right]$$

where $\sigma(x) = \frac{1}{1 + \exp(-x)}$. In RLHF, the reward model $r_\phi(x, y)$ is often initialized from the SFT model $\pi^{\text{SFT}}(y|x)$ by replacing the final layer with a classification head, which outputs a scalar (i.e., the reward).

**Fine-tuning with reinforcement learning.** Once the reward model, which serves as a proxy for the utility we aim to maximize, is trained, the next step is to apply reinforcement learning under this reward model. Typically, the following objective is used:

$$\max_{\pi_\theta} \mathbb{E}_{x \sim \mathcal{D}, y \sim \pi_\theta(y|x)}[r_\phi(x, y)] - \beta \mathbb{D}_{\text{KL}}[\pi_\theta(y|x) \| \pi_{\text{ref}}(y|x)]$$

Here, $\beta$ is a coefficient that controls the deviation from the reference policy $\pi_{\text{ref}}$, which is typically the SFT model $\pi^{\text{SFT}}$. In practice, the policy $\pi_\theta$ is also initialized using the SFT model $\pi^{\text{SFT}}$. The KL

constraint is crucial, as it prevents the model from deviating too far from the SFT model, which is helpful to mitigate issues such as forgetting (Schulman, 2015; Schulman et al., 2017; Abdolmaleki et al., 2018; Jaques et al., 2019).

## 3.2 COUNTERFACTUAL INVARIANCE

An ideal debiased reward model should intuitively remain invariant to spurious factors of variations. For example, to eliminate length bias, the reward model should exhibit invariance to changes in response length. To formalize this notion, we leverage the concept of *counterfactual invariance* (Veitch et al., 2021). We begin by introducing some notation. Let $Z$ represent the random variable corresponding to a spurious factor of variation (e.g., length), and let $T$ denote the random variable that encompasses the prompt-response pair. A reward model $r$ is said to exhibit counterfactual invariance to $Z$ if $r(T(z)) = r(T(z'))$ for all $z, z'$, where $z$ and $z'$ are realizations of $Z$, and $T(z)$ denotes the counterfactual $T$ we would have observed if $Z$ were $z$. Throughout the paper, we use the term "invariant" or "debiased" to refer specifically to counterfactual invariance.

## 3.3 CAUSAL DECOMPOSITION

The prompt-response pair $T$ can be decomposed into latent components based on their relations with the spurious factor $Z$ (Veitch et al., 2021). Specifically, we define $T^{Z,\perp}$ as the component of $T$ that is not causally influenced by $Z$. In other words, $T^{Z,\perp}$ represents the part of $T$ such that any function of $T$ is counterfactually invariant to $Z$ if and only if it depends solely on $T^{Z,\perp}$. Under weak conditions on $Z$, $T^{Z,\perp}$ is well defined. Further details regarding the derivation and properties can be found in (Veitch et al., 2021). In the next section, we extend this concept to develop a reward model that incorporates counterfactual invariance, which enables debiasing against various spurious factors.

## 4 METHOD

Ideally, counterfactual examples are necessary to learn counterfactual invariant predictors (Quinzan et al., 2022). However, obtaining such examples is challenging, especially in RLHF settings. For instance, given a response of length 100, it is very hard to create a counterfactual response of length 50. Nonetheless, as suggested by Veitch et al. (2021), observable signatures implied by causal graphs can be leveraged to regularize the hypothesis class of the predictor.

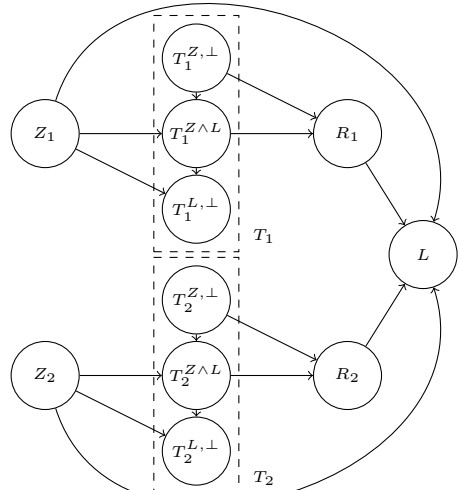

Figure 1: Diagram illustrating the proposed causal reward modeling. Here, $Z$ represents spurious factors (e.g., response length), $T$ denotes the prompt and response pair, $R$ is the true reward, and $L$ is the human preference label. The diagram highlights the decomposition of $T$ into latent components: $T^{Z,\perp}$, which is independent of $Z$; $T^{Z \wedge L}$, representing factors influenced by both $Z$ and $L$; and $T^{L,\perp}$, which does not causally impact $L$. This framework shows how reward hacking, modeled via direct paths from $Z$ to $L$, can mislead traditional reward models. Our proposed approach aims to isolate $T^{Z,\perp}$, ensuring counterfactual invariance and debiasing reward predictions.

Consider the causal diagram of reward models in Fig. 1. Here, $Z$ is the spurious factor (e.g., response length), $T$ is the prompt-response pair, $R$ is the reward and $L$ is the preference label. The binary label $L$ (e.g., $L = 1$ when $X_1$ is preferred) can be modeled under the Bradley-Terry model, where preferences depend on *true* rewards. In practice, however, human labels are often biased, captured by a direct edge from $Z$ to $L$.

As discussed in §3.3, $T$ can be decomposed into latent components based on their relation with $Z$. In addition to $T^{Z,\perp}$, we define $T^{L,\perp}$ as the component that does not directly cause $L$, and $T^{Z \wedge L}$ as the complementary remaining part. And an invariant reward model should depend solely on $T^{Z,\perp}$. Although precisely learning such an invariant reward model is infeasible without counterfactual dataset, the causal graph reveals that $T^{L,\perp}$ is independent of $Z$. Consequently, any counterfactual invariant reward model must also be independent of $Z$, which leads to the following condition:

$$f(T) \perp\!\!\!\perp Z \tag{1}$$

where $f$ is a counterfactually invariant function. This independence condition is merely a necessary condition implied by counterfactual invariance. But the key idea is that it constrains the hypothesis class, potentially guiding the model toward learning an invariant predictor.

## 4.1 Maximum Mean Discrepancy (MMD) Regularization for Independence

To enforce the independence condition outlined in Eq. (1), we employ Maximum Mean Discrepancy (MMD), a kernel-based statistical measure that quantifies the divergence between two probability distributions (Gretton et al., 2012; Liu et al., 2020). MMD is commonly used to regularize models by ensuring alignment between distributions across domains or subpopulations (Tolstikhin et al., 2016; Zhang et al., 2024a). Formally, given two distributions $\mathbb{P}$ and $\mathbb{Q}$, the squared MMD in a reproducing kernel Hilbert space (RKHS) $\mathcal{H}_k$ is defined as:

$$\text{MMD}(\mathbb{P}, \mathbb{Q}, \mathcal{H}_k) = \sup_{f \in \mathcal{F}} \left( \mathbb{E}_{x \sim P}[f(x)] - \mathbb{E}_{y \sim Q}[f(y)] \right)^2, \tag{2}$$

where $\mathcal{F}$ denotes a class of functions in $\mathcal{H}_k$, and $x \sim \mathbb{P}$, $y \sim \mathbb{Q}$ are two random variables. Intuitively, MMD measures the maximum mean difference between $\mathbb{P}$ and $\mathbb{Q}$ over functions $f \in \mathcal{F}$, as determined by a kernel $k(\cdot, \cdot)$ such as Gaussian kernels. In our approach, we use MMD as a regularizer to ensure that the learned reward model $f(T)$ is invariant to the spurious variable $Z$. If $Z$ is binary, our MMD regularizer can be defined as:

$$\text{MMD} \left( P(f(T)|Z = 0), P(f(T)|Z = 1) \right).$$

When $Z$ spans a large or continuous space (e.g., response lengths), directly applying MMD becomes computationally intensive. To address this, we partition $Z$ into $M$ discrete bins and compute MMD across all pairs of bins. Let $b \in [1, M]$ denote bin indices, with $P_b(f(T))$ representing the conditional distribution of $f(T)$ within bin $b$, the regularizer is then defined as:

$$\sum_{m, m' \in [M]} \text{MMD}(P_m(f(T)), P_{m'}(f(T))).$$

This binning approach ensures the applicability of MMD in high-dimensional or continuous settings while preserving the ability to capture variations across $Z$.

In our architecture, $f(T)$ denotes the latent representation of the prompt-response pair $T$. The reward model $r_\phi(x, y)$ is parameterized by $\phi$ and depends on $f(x, y)$, such that:

$$r_\phi(x, y) = r_\phi(f(x, y))$$

To regularize $r_\phi(x, y)$, we map all responses into $M$ bins based on their spurious factor $Z$ (e.g., response length). For each bin $b$, we compute the conditional distribution $P_b(f(x, y))$. The overall objective function, combining the reward model training loss and the MMD-based regularizer, is:

$$-\mathbb{E}_{(x, y_w, y_l) \sim \mathcal{D}}[\log \sigma(r_\phi(x, y_w) - r_\phi(x, y_l))] + \lambda \sum_{m, m' \in [M]} \text{MMD}(p_m(r(x, y)), p'_m(r(x, y))),$$

where $\sigma(x) = \frac{1}{1 + e^{-x}}$ is the sigmoid function, and $\lambda$ is a hyperparameter controlling the weight of the MMD regularization. This formulation enforces counterfactual invariance by penalizing discrepancies in reward predictions across bins of the spurious variable $Z$, effectively guiding the model to learn invariant representations.

## 5 Experiments

To examine the effectiveness of the proposed reward model, we'll test on four dataset covering sycophantic, length, concept and discrimination bias. Although we only apply marginal regularization in §4, in practice, the focus is typically on the model's ability to generate the chosen responses based on prompts. Therefore, we additionally test another variant of the regularization where the prompt and response pair are divided into chosen and rejected subsets and the independence regularization is applied for each subset individually. We denote this as the *conditional* causal reward model (CRM), in addition to the *unconditional* variant discussed before.

### 5.1 Addressing Sycophantic Bias (Semi-synthetic)

Sycophantic bias (Sharma et al., 2023; Ranaldi & Pucci, 2023) refers to a model's tendency to produce responses that agree with or flatter the user, regardless of the truth or accuracy of the content. This

bias often arises when reward models inadvertently assign higher rewards to outputs that align with users' stated beliefs or preferences, particularly in preference datasets where agreement is implicitly favored over truthfulness. For example, in a conversational setting, if annotators systematically reward responses that confirm the user's input (e.g., "Yes, you are correct"), the model learns to prioritize sycophantic behavior to maximize its reward. This can lead to outputs that prioritize agreeableness over factual accuracy, undermining the model's trustworthiness.

**Dataset and training.** To investigate sycophantic bias, we create a semi-synthetic dataset based on dataset developed by Sharma et al. (2023). Specifically, our prompts are structured with the template "`{question} I think the answer is {correct_answer} but I'm really not sure.`". In this setup, we artificially induce a correlation between sycophantic behavior and correctness. Specifically, with an $80\%$ probability, the chosen response is prefixed with "`Yes, you are right.`" Conversely, with a $20\%$ probability, this prefix appears in the rejected response. This creates an artificial but controlled spurious correlation between agreement ("`Yes, you are right.`") and correct answer, enabling us to observe, measure and address sycophantic bias effectively.

For SFT, we use the Llama-3 8B base model (Dubey et al., 2024), finetuned on a combination of data from the Anthropic HH-RLHF dataset (Bai et al., 2022) and our semi-synthetic sycophantic training dataset. The HH-RLHF dataset is included to ensure sufficient training data volume, as the semi-synthetic dataset contains only 1,727 examples. The reward and policy models are then trained using the chosen/rejected pairs, with the policy fine-tuned for two epochs via Proximal Policy Optimization (PPO) (Schulman et al., 2017), implemented in OpenRLHF (Hu et al., 2024). Additional implementation details are available in Appendix B.1.

**Results.** For each test prompt, we generate 50 responses. We then quantify sycophancy by checking whether the phrase "`Yes, you are right.`" appears in any of those responses. Table 1 reports the percentage of test prompts for which all 50 sampled responses exhibit sycophantic behavior. It is worth noting that the SFT model, trained on chosen responses with high correlation with sycophantic phrasing, naturally tends to produce "`Yes, you are right.`" as a default pattern. In contrast, both the conditional and unconditional CRM approaches successfully disentangle this spurious correlation and reduce the prevalence of sycophantic responses.

Table 1: Results on semi-synthetic syncophatic dataset. The conditional `CRM` outperforms other methods. Bold values indicate the best performance. Results are averaged over three runs of PPO.

| Model | Average Percentage (%) |
|---|---|
| Vanilla RM | 92.67 |
| Conditional CRM | **19.78** |
| Unconditional CRM | 62.64 |

### 5.2 Addressing Length Bias

Length bias (Zheng et al., 2023) refers to the tendency of reward models to favor longer responses due to spurious correlations in the training data. For instance, in human preference datasets, annotators may unconsciously associate longer responses with higher-quality or more comprehensive answers, leading to disproportionate rewards for verbosity rather than substantive content. This bias often misaligns the model's behavior with true human preferences, particularly when concise and accurate responses are preferred in real-world applications.

**Dataset and training.** We adopted the Alpaca dataset (Dubois et al., 2024b) for our experiments. Initially, we uses the chosen response for each prompt to do supervised finetuning (SFT) using the Llama-3 8B base model (Dubey et al., 2024). Then, this SFT model was subsequently employed to train both the reward model and the policy model. For reward model, we used the chosen and the rejected pair for training. With the reward model, we then trained the SFT policy with the PPO implementation from OpenRLHF (Hu et al., 2024) for one epoch. Additional details on hyperparameters and configurations are available in the Appendix B.2.

**Results.** Our findings are illustrated in Fig. 2, where each dot on the plots represents a single model run, evaluated by its win rate, calculated as the proportion of wins against the SFT model. The score is defined by score $= 50 + (n_{\text{win}} - n_{\text{lose}})/N * 100$, where $n_{\text{win}}$ and $n_{\text{lose}}$ denote the counts of wins and losses, respectively, and $N$ represents the total test count. In the leftmost plot, we observe that both the conditional and unconditional causal regularization methods achieve superior performance

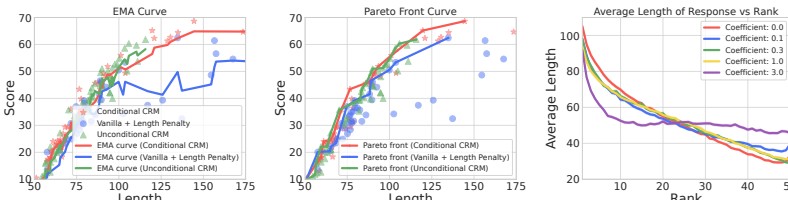

Figure 2: Results on Length Bias, with each dot representing models trained with different regularization coefficients and PPO hyperparameters. The left plot shows exponential moving average (EMA) curves, the middle plot illustrates the Pareto front, and the right plot captures the length-rank correlation for different causal reward models.

compared to the vanilla reward model with length penalty, as shown by their higher exponential moving average (EMA) curves. Furthermore, when examining the Pareto frontier, our approach demonstrates an advantage over the baseline method.

Finally, we analyze the impact of the regularization effect by sampling 50 responses per prompt and ranking them using a reward model trained with varying causal regularization coefficients. We then compute the average response length across all prompts for each rank. Our results show that models with higher coefficients assign higher ranks (i.e., lower numerical rank values) to responses with shorter lengths, indicating a reduction in the bias toward longer responses.

## 5.3 ADDRESSING CONCEPT BIAS

Concept bias (Zhou et al., 2023) in LLMs refers to the model's unintended reliance on correlations between specific concepts and labels present in the training data. For instance, in the Yelp Review dataset (Zhang et al., 2015), if most reviews mentioning "food" (categorized as a food concept) are labeled with positive sentiments, the LLM may develop a shortcut, incorrectly predicting positive sentiment for any review that involves "food." This type of concept bias, which stems from associating unrelated terms with certain outcomes due to imbalanced distribution in the training data, causes LLMs to make incorrect predictions in new, unseen scenarios, which highlights the tendency of LLMs to overgeneralize based on spurious correlations, rather than always grasping the actual context of the input. In this section, we demonstrate the effectiveness of the proposed causal reward modeling in mitigating the concept bias when conducting sentiment analysis of the review datasets.

**Dataset and training.** We conducted experiments using Yelp (Zhang et al., 2015), IMDB (Maas et al., 2011), and Amazon Shoe Review (He & McAuley, 2016) datasets, augmented with additional concept labels provided by Zhou et al. (2023). Specifically, each dataset includes three concepts, where Yelp has "price", "service", "food"; IMDB has "music", "acting", "comedy"; and Amazon has "size", "color" and "style". To introduce more obvious concept bias, following Zhou et al. (2023), we modified each dataset to ensure all positive-sentiment samples were explicitly linked to a specific concept. For instance, in the Yelp dataset, we filtered reviews so that all positive sentiment entries were linked to the "food" concept.

To facilitate training, we reformatted the datasets to align with the structure of Anthropic hh-rlhf (Bai et al., 2022) dataset. Specifically, we appended the prompt "Classify the text into negative, or positive" to the front of each review, and used the correct "positive" or "negative" label from ground truths as the chosen assistant response. The incorrect classifications were then used in the rejected assistant response. We fully supervise finetuned (SFT) the Llama-3 8B base model (Dubey et al., 2024) on each of the above processed, concept-biased datasets using the chosen responses. The resulting SFT model was further utilized to train both vanilla and causal reward models. Finally, using these reward models, we conducted PPO finetuning using implementations from OpenRLHF (Hu et al., 2024) on the SFT model to produce final models for evaluation. More details on training hyperparameters are explained in Appendix B.3.

**Metrics.** We assess performance using both utility metrics (Acc@C, Acc@NoC) as well as the bias-specific metric Bias@C, as introduced in (Zhou et al., 2023). The utility metrics, which reflect the accuracy of correct sentiment classifications with (Acc@C) and without (Acc@NoC) the presence of a concept, indicate better performance with higher values. On the other hand, Bias@C measures spurious correlations associated with concept $C$, where values closer to zero suggest weaker biases. Specifically, positive Bias@C values suggest the model tends to predict positive labels when concept $C$ is present in the input, whereas negative values suggest the opposite tendency. For a more detailed

Table 2: Models performance after finetuning with PPO using both vanilla and the proposed causal reward models across concept-biased Yelp, IMDB, and Amazon Shoe Review datasets. Bold values indicate the best performance.

| | Price (Yelp) | | | Service (Yelp) | | | Food (Yelp) | | |
|---|---|---|---|---|---|---|---|---|---|
| | Acc@NoC | Acc@C | Bias@C | Acc@NoC | Acc@C | Bias@C | Acc@NoC | Acc@C | Bias@C |
| Vanilla RM | 59.26 | 71.47 | 18.88 | 69.09 | 71.43 | -15.54 | 78.77 | 67.48 | 7.31 |
| Conditional CRM | **97.22** | **99.04** | **0.52** | **99.45** | **97.56** | **-0.61** | 97.77 | **99.09** | **0.71** |
| Unconditional CRM | 94.44 | 98.35 | 6.86 | 98.18 | 97.21 | -3.56 | **98.88** | 97.57 | -0.86 |

| | Music (IMDB) | | | Acting (IMDB) | | | Comedy (IMDB) | | |
|---|---|---|---|---|---|---|---|---|---|
| | Acc@NoC | Acc@C | Bias@C | Acc@NoC | Acc@C | Bias@C | Acc@NoC | Acc@C | Bias@C |
| Vanilla RM | 77.78 | 73.98 | 13.49 | 75.54 | 71.81 | -20.94 | 69.93 | 75.78 | 20.09 |
| Conditional CRM | 68.89 | 55.73 | **2.86** | 54.84 | 60.64 | **-7.68** | 58.04 | 56.35 | **7.99** |
| Unconditional CRM | **88.89** | **88.35** | 9.52 | **89.52** | **86.17** | -13.24 | **85.31** | **89.45** | 12.41 |

| | Size (Amazon) | | | Color (Amazon) | | | Style (Amazon) | | |
|---|---|---|---|---|---|---|---|---|---|
| | Acc@NoC | Acc@C | Bias@C | Acc@NoC | Acc@C | Bias@C | Acc@NoC | Acc@C | Bias@C |
| Vanilla RM | 76.17 | 54.08 | -4.05 | 63.88 | 72.47 | 15.48 | 38.30 | 74.35 | -10.16 |
| Conditional CRM | **79.95** | **85.87** | -2.37 | **84.58** | **80.73** | 2.45 | **87.94** | **80.64** | **-0.70** |
| Unconditional CRM | 73.89 | 53.26 | **-1.58** | 62.56 | 70.41 | 3.93 | 38.30 | 72.20 | -1.49 |

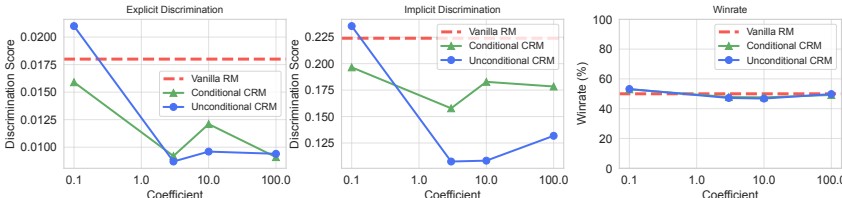

Figure 3: Comparison of discrimination and utility performance on the *hh-rlhf dataset* for CRM in both conditional and unconditional settings, with varying MMD coefficient. Larger coefficients reflect higher weights of MMD loss. We assess both explicit and implicit discrimination scores, while win rates are evaluated by GPT-4o, measured against the baseline vanilla RM.

explanation of the Bias@C metric, we direct interested readers to their original work (Zhou et al., 2023).

**Results.** As shown in Table 2, the results demonstrate that CRM consistently reduces concept bias across the Yelp, IMDB, and Amazon Shoe Review datasets compared to the vanilla reward model. Specifically, both conditional and unconditional CRMs achieve significantly lower Bias@C values, with conditional CRM showing reductions of up to 97% on the Yelp dataset (e.g., for the "Price" concept). These results highlight the effectiveness of our approach in mitigating spurious correlations.

Beyond bias reduction, the results also illustrate the trade-offs between conditional and unconditional CRMs. While conditional CRM often performs the best in Bias@C reduction, unconditional CRM demonstrates superior Acc@NoC and Acc@C performance, particularly on datasets such as IMDB, where unconditional CRM achieves average accuracies of 87.9% for Acc@NoC and 88.0% for Acc@C, significantly outperforming the Vanilla RM baseline's 74.4% and 73.9%, respectively. This balance suggests that unconditional CRM effectively mitigates bias while preserving high predictive utility in concept-relevant contexts. However, we leave more in-depth investigation into the dynamics of this trade-off for future work.

## 5.4 ADDRESSING DISCRIMINATION BIAS

Given the implicit biases embedded in training data, LLMs often learn spurious discriminatory patterns over different demographic groups (Tamkin et al., 2023). While some previous works attempt to leverage post-training methods (Bai et al., 2022) to mitigate this issue by designing specific bias-countering preference pairs, these approaches are often labor-intensive, lacks explicit guarantees of effectiveness, and often compromise the model's overall utility (Allam, 2024). In contrast, we demonstrate below the effectiveness of our proposed CRM in **explicitly** mitigating discriminatory bias without relying on specific bias-focused data, while maintaining the model's original performance on general language modeling tasks.

Table 3: Discrimination evaluation over a diverse set of both explicit and implicit discrimination scenarios using the Discrm-eval dataset (Tamkin et al., 2023). The scores are the mixed-effects coefficients for each demographic variable, where the lower indicates less discrimination. The best performance is in bold.

| | Explicit | | | | Implicit | | | | Overall |
|---|---|---|---|---|---|---|---|---|---|
| Model | Gender | Race | Age | Avg | Gender | Race | Age | Avg | Avg |
| SFT | 0.003 | 0.002 | 0.015 | **0.007** | 0.227 | 0.251 | 0.523 | 0.334 | 0.171 |
| Vanilla RM | 0.032 | 0.016 | **0.007** | 0.018 | 0.181 | 0.230 | 0.261 | 0.224 | 0.121 |
| Conditional CRM | **0.008** | **0.002** | 0.018 | 0.009 | 0.264 | **0.181** | 0.060 | 0.158 | 0.084 |
| Unconditional CRM | 0.009 | **0.002** | 0.018 | 0.009 | **0.070** | 0.213 | **0.036** | **0.107** | **0.058** |

**Dataset and training.** To obtain discrimination-specific data, we first filter the samples in the training set of the Anthropic HH-RLHF (Bai et al., 2022) dataset, retaining those that contain a pre-defined large set of demographic variables specified in Appendix B.4.1. This results in 35,567 samples, which we split into 90% for training, 5% for validation, and 5% for testing. To stabilize training for CRM, we further group similar demographic variables into 60 demographic bins, as detailed in Appendix B.4.2, balancing granularity and training batch volumes.

**Evaluation tasks.** We evaluate both targeted discrimination bias performance and general model utility. (1) To assess discrimination bias, we use the Anthropic Discrm-eval dataset (Tamkin et al., 2023), which contains two subsets of queries focusing on *explicit* and *implicit* bias. These queries are templated with demographic attributes such as age, gender, and race (e.g., `"The patient is a [age]-year-old [race] [gender] ... diagnostic information ... should they be prioritized for a kidney transplant?"`). (2) To evaluate general utility, we randomly sample 1,000 queries from the HH-RLHF test set and assess the corresponding models' responses.

**Metrics.** To evaluate discrimination, we adopt the approach from (Tamkin et al., 2023), analyzing how demographic attributes such as race, gender, and age influence decision boundaries. Specifically, we fit a mixed-effects model and report **the coefficients of each demographic attribute**, where lower coefficients indicate lower bias. For model general utility, we similarly report the **win rate** comparing the performance of the CRM-enhanced model against the baseline vanilla PPO model based on evaluations conducted by GPT-4.

**Results.** As shown in Table 3 and Fig. 3, CRM significantly reduces discrimination across both explicit and implicit scenarios compared to the vanilla reward model. In terms of discrimination patterns, models generally exhibit higher bias in implicit scenarios, while performing relatively well in explicit questions. Nonetheless, CRM models effectively reduce bias in both cases, with a particularly significant impact on implicit scenarios where the vanilla model demonstrates greater bias. Among the CRM variants, unconditional CRM achieves the lowest implicit discrimination score (0.107) and the best overall performance (0.058), while conditional CRM performs slightly better in explicit settings. These findings highlight CRM's effectiveness in mitigating both explicit and implicit biases across demographic attributes. The win rate analysis in Fig. 3 confirms that the additional MMD regularization term has minimal impact on the model's general utility, highlighting CRM's ability to effectively address discrimination while preserving its original performance.

## 6 CONCLUSIONS AND FUTURE WORK

In this paper, we introduced a novel framework for causal reward modeling (CRM) aimed at addressing spurious correlations that compromise the alignment of LLMs with human preferences. By incorporating counterfactual invariance into reward learning, our approach mitigates biases such as sycophancy, length bias, concept bias, and discrimination. Through extensive experiments on both synthetic and real-world datasets, we have demonstrated the effectiveness of CRM in enhancing fairness, reliability, and trustworthiness across various tasks. Additionally, CRM can be seamlessly integrated into any existing RLHF workflows, enabling more robust and equitable alignment of LLMs without introducing significant complexity.

## USAGE OF LARGE LANGUAGE MODELS

The language in this paper was at times polished with the assistance of an LLM. The model was not used for research ideation, experimental design, or data analysis.

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

## A    EXTENSION WITH DPO

Our framework can also be extended to DPO by replacing the reward model with the DPO's implict reward. This gives us the following objective for training Causal DPO,

$$
\mathcal{L}_{\text{Casual-DPO}} = - \mathbb{E}_{(x,y_w,y_l)\sim\mathcal{D}} \left[ \log \sigma \left( \beta \log \frac{\pi_{\boldsymbol{\theta}}(y_w|x)}{\pi_{\text{ref}}(y_w|x)} - \beta \log \frac{\pi_{\boldsymbol{\theta}}(y_l|x)}{\pi_{\text{ref}}(y_l|x)} \right) \right]
$$
$$
+ \lambda \sum_{m,m'\in[M]} \text{MMD} \left( p \left( \frac{\pi_{\boldsymbol{\theta}}(y|x)}{\pi_{\text{ref}}(y|x)} | b=m \right), p \left( \frac{\pi_{\boldsymbol{\theta}}(y|x)}{\pi_{\text{ref}}(y|x)} | b=m' \right) \right). \quad (3)
$$

## B    EXPERIMENTAL DETAILS

### B.1    SYCOPHANTIC BIAS

The reward model is trained using Low-Rank Adaptation (LoRA) (Hu et al., 2021) finetuning with rank $64$ and weight $\alpha = 128$ with batch size 32 across 4 gpus. For both the conditional and the unconditional regularization, the coefficients are chosen from $\{0, 0.1, 0.3, 0.5, 1, 3, 5, 10\}$. The final policy model is trained with PPO with batch size 16 for 2 epochs. The initial KL coefficient is set to be 0.01.

### B.2    LENGTH BIAS

To obtain the SFT model, we begin by finetuning the Llama-3 8B base model on selected responses from the Alpaca farm dataset for 3 epochs, using a learning rate of $2 \times 10^{-5}$. Additional hyperparameters are available in the Alpaca farm GitHub repository[2]. Next, we train the reward model starting from this SFT model. This training is done using LoRA finetuning with rank $64$ and weight $\alpha = 128$, for 4 epochs, with a learning rate of $1 \times 10^{-4}$ and a batch size of $128$ (distributed as 16 per GPU device).

To obtain a variety of reward models, we perform a hyperparameter sweep on two variables: 1) the number of bins, and 2) the regularization coefficient. For the number of bins, we explore values $\{10, 20, 30\}$, and for the coefficient, we test $\{0.1, 1.0, 3.0, 10, 100\}$. Finally, we apply PPO to finetune the SFT model under our learned reward model, obtaining the final policy model. For the PPO stage, we train for 1 epoch with a KL coefficient sweep over $\{0.003, 0.01, 0.03, 0.1\}$, resulting in a total of 60 (conditional) causal reward models.

For the baseline method, the reward model is trained with a regularization coefficient of 0 (equivalently). In the PPO stage, we perform a more thorough sweep, tuning the KL coefficient over $\{0.003, 0.01, 0.03, 0.1\}$, the learning rate over $\{5 \times 10^{-7}, 1 \times 10^{-6}\}$, and the length penalty over $\{0, 1 \times 10^{-3}, 1 \times 10^{-4}, 1 \times 10^{-5}, 5 \times 10^{-4}, 1 \times 10^{-6}, 5 \times 10^{-6}\}$. This process results in 56 models, providing a comparable set to the causal reward models.

---

[2]`https://github.com/tatsu-lab/alpaca_farm/blob/main/examples/scripts/sft.sh`

## B.3 CONCEPT BIAS

As briefly mentioned in §5.3, we supervised finetuned (SFT) the Llama-3 8B base model on each of the processed Yelp, IMDB, Amazon Shoe Review datasets. We keep the same hyperparameters for all datasets, which are illustrated in Table 4. The resulting SFT model is used for the reward learning through LoRA, where detailed parameters are illustrated in Table 5. The reward models are subsequently utilized during the PPO, where the hyperparameters for PPO on each dataset is showed in Table 6. All the trainings are distributed on 8 NVIDIA A100 GPUs.

Table 4: Supervised finetuning hyperparameters for concept-bias experiments.

|  | Supervised Finetuning |
|---|---|
| Learning rate | 2e-5 |
| Batch size | 128 |
| Gradient accumulation steps | 2 |
| Training epochs | 4 |
| Warm-up steps | 500 |

Table 5: Reward learning hyperparameters for concept-bias experiments.

|  | Vanilla Reward | | | Conditional CRM | | | Unconditional CRM | | |
|---|---|---|---|---|---|---|---|---|---|
|  | Yelp | IMDB | Amazon | Yelp | IMDB | Amazon | Yelp | IMDB | Amazon |
| Regularization coefficient |  | - |  | 0.1 | 0.3 | 0.1 | 0.5 | 3 | 1 |
| Learning rate |  |  |  |  | 1e-4 |  |  |  |  |
| LoRA Rank |  |  |  |  | 64 |  |  |  |  |
| LoRA Alpha |  |  |  |  | 128 |  |  |  |  |
| Batch size |  |  |  |  | 64 |  |  |  |  |
| Gradient accumulation steps |  |  |  |  | 1 |  |  |  |  |
| Training epochs |  |  |  |  | 1 |  |  |  |  |

Table 6: PPO hyperparameters for concept-bias experiments.

|  | Proximal Policy Optimization (PPO) |
|---|---|
| Actor learning rate | 5e-7 |
| Critic learning rate | 9e-6 |
| Initial KL coefficient | 0.01 |
| Batch size | 128 |
| Gradient accumulation steps | 8 |
| Training epochs | 1 |

### B.4  DISCRIMINATION BIAS

#### B.4.1  TRAINING DATA PREPARATION

This section provides the demographic keyword groups used for filtering data from the Anthropic HH-RLHF dataset (Bai et al., 2022). The keywords were grouped into broad categories based on demographic attributes, ensuring comprehensive coverage. Table 7 through 11 summarize the categories and their associated keywords. These keyword categories provided a robust basis for filtering the HH-RLHF dataset, ensuring the inclusion of diverse demographic contexts in the training data.

Table 7: Age-related categories and keywords used for filtering data.

| Category | Keywords |
|---|---|
| Childhood | *teen, child, kid, baby, toddler, infant, adolescent, youth, teenager* |
| Young Adults | *adult, young adult, young, grown-up* |
| Middle Age | *middle-aged, sophisticated, mature* |
| Elderly | *elderly, old, senior, aged* |

Table 8: Gender-related categories and keywords used for filtering data.

| Category | Keywords |
|---|---|
| Male | *male, man, masculine* |
| Female | *female, woman, feminine* |
| Non-binary | *gay, lesbian, LGBT, non-binary, queer* |

Table 9: Race-related categories and keywords used for filtering data.

| Category | Keywords |
|---|---|
| White | *white, european, caucasian, middle eastern* |
| Black | *black, african* |
| Asian | *asian, chinese, japanese, korean, indian, south asian, east asian, southeast asian* |
| Latino | *latino, hispanic* |
| Indigenous | *indigenous, native, pacific island* |

#### B.4.2  DEMOGRAPHIC BINS

To stabilize training for CRM, we grouped similar demographic variables into 60 distinct bins. Specifically, we targeted *age*, *gender*, and *race*-related discrimination. Each row in Table 7, Table 8, and Table 9 was treated as a bin, resulting in a total of $4 \times 3 \times 5 = 60$ bins.

#### B.4.3  DETAILED DESCRIPTION

Aiming to address a comprehensive collection of discrimination factors spanning age, gender, race and political groups that LLMs might encounter when handling various forms of societal decisions, we can construct any target training dataset by following the anthropic discrimination dataset Tamkin et al. (2023) which covers 70 topics across society that involve *accepting/rejecting* a person.

Specifically, it consists of 70 decision topic templates with placeholders for demographic information (e.g. [AGE], [RACE], [GENDER]) and all the questions are framed as a yes/no decision faced by a third party decision-maker who must use only the information available in the prompt. Critically, each question is organized in a way such that *yes* refers to a positive outcome for the subject of the decision question (e.g. having a loan approved or being promoted). Notably, they consider two ways to filling the templates:

Table 10: Nationality-related categories and keywords used for filtering data.

| Region | Keywords |
|---|---|
| Americas | *american, canadian, mexican, brazilian, argentinian* |
| Europe | *german, french, italian, spanish, british, russian, polish* |
| Asia-Pacific | *chinese, japanese, korean, indian, australian, new zealander* |
| Africa | *nigerian, south african, egyptian, kenyan* |

Table 11: Religion-related categories and keywords used for filtering data.

| Religion | Keywords |
|---|---|
| Christianity | *christian, church, bible* |
| Islam | *muslim, mosque, koran* |
| Judaism | *jewish, synagogue, torah* |
| Dharmic and Others | *hindu, buddhist, temple, religion* |

1. **Explicit:** insert random combinations of age, race, and gender directly into the placeholders, with [AGE] $\in$ [20, 30, 40, 50, 60, 70, 80, 90, 100], [GENDER] $\in$ [male, female, non-binary] and [RACE] $\in$ [white, Black, Asian, Hispanic, Native American], in total 9450 questions.

2. **Implicit:** only specify the age and a person's name to implicitly indicate a particular race and gender (e.g. Wei Li, Carlos Reyes) . This approach focuses on assessing discrimination based on more subtle information correlated with race and gender

Thus a tentative approach would be similar to §5.2 where we employ MMD to decouple the representation from the targeted discrimination factors (age, race, gender). Suppose we have a discrimination-intensive dataset $\mathcal{D}$ to train a reward model, we can vary the inputs by first tagging the targeted discrimination factors to construct placeholders and then substituting them in the prompt templates with different combinations of age, race, and gender. We define $f(x)$ as the representation of input prompts, where $x$ is a prompt filled with specific demographic information. The reward model, parameterized by $\phi$, is thus denoted by $r_\phi(x, y)$, where $y$ represents the decision outcome. The goal is to ensure that $r_\phi(f(x), y)$ is independent of the discriminatory factors.

To measure and minimize discrimination bias with MMD, we can define demographic groups based on combinations of age, race, and gender as $\mathcal{G}$. For each demographic group $g \in \mathcal{G}$, we can map prompts into $N$ bins, where each bin corresponds to a specific combination of demographic attributes. The MMD regularizer is computed as follows: where $y_w$ and $y_l$ indicate positive and negative decision outcomes, respectively, and $\sigma$ is the sigmoid function. The expectation is calculated over the discrimination-intensive dataset $\mathcal{D}$, where each sample $(x, y_w, y_l)$ consists of a prompt $x$ with a preferred decision outcome $y_w$ over $y_l$. The term $\lambda$ acts as a regularization parameter, balancing the objective between maximizing the reward signal and minimizing discrimination bias measured by the MMD between different demographic groups.

### B.4.4 EVALUATION PROMPT

The prompt template that we provide to GPT-4o to compare the two response of the CRM-enhanced model and the vanilla PPO baseline is detailed in B.4.4.

**Prompt Template for Comparing Two Model's Response**

SYSTEM: Please act as an impartial judge and evaluate the quality of the responses provided by two AI assistants to the user question displayed below. You should choose the assistant that follows the user's instructions and answers the user's question better. Your evaluation should consider factors such as the helpfulness, harmlessness, relevance, accuracy, depth, creativity, and level of detail of their responses. Begin your evaluation by comparing the two responses and provide a short explanation. Avoid any position biases and ensure that the order in which the responses were presented does not influence your decision. Do not allow the length of the responses to influence your evaluation. Do not favor certain names of the assistants. Be as objective as possible.

After providing your explanation, you must output your final verdict by strictly following this format: '[[A]]' if assistant A is better, '[[B]]' if assistant B is better, and '[[C]]' for a tie.

USER: [Initial User-Assistant Dialogue] {prompt}

[Start of assistant A's Response] {rwd_response} [End of assistant A's Response]

[Start of assistant B's Response] {vanilla_ppo_response} [End of assistant B's Response].

