# OpenReview forum: "Beyond Reward Hacking: Causal Rewards for Large Language Model Alignment"
_ICLR.cc/2026/Conference — Submitted to ICLR 2026_

### Official Review · Reviewer_Dqv3 · 2025-10-29

**Soundness:** 3
**Presentation:** 3
**Contribution:** 3
**Rating:** 8
**Confidence:** 3

**Summary:**

The paper addresses a fundamental problem in Reinforcement Learning from Human Feedback (RLHF) — reward hacking due to spurious correlations in reward modeling. Traditional RLHF reward models often conflate superficial correlations (e.g., response length, agreement, demographic cues) with genuine human preferences, leading to biases such as length bias, sycophancy, concept bias, and discrimination bias (see Section 1). To overcome this, the authors propose a Causal Reward Model (CRM) that incorporates counterfactual invariance to ensure that reward predictions are independent of irrelevant variables (Section 3.2–4.1). The method employs Maximum Mean Discrepancy (MMD) regularization to enforce statistical independence between latent representations and spurious factors (Eq. (1), Section 4.1). Experiments across four settings — sycophantic, length, concept, and discrimination bias — demonstrate that CRM significantly mitigates spurious correlations and improves fairness and robustness without sacrificing model utility (Sections 5.1–5.4, Tables 1–3, Fig. 3). The approach integrates seamlessly with existing RLHF pipelines, offering a practical improvement to current alignment workflows.

**Strengths:**

Originality: The paper introduces a novel causal regularization framework for reward modeling within RLHF, which is an underexplored direction. While prior works address specific biases (e.g., length bias via PoE or WARM), CRM provides a unified causal formulation that generalizes across multiple types of spurious correlations (Section 2.2). The concept of counterfactual invariance adapted from causal inference to LLM alignment is new.

Quality: The theoretical foundation is clearly established with causal diagrams (Figure 1, Section 3.2) and well-defined objectives (Eq. (1)–(2), Section 4.1). The experimental design is comprehensive and evaluates diverse bias types. Results (e.g., Table 2 and 3) consistently demonstrate the superiority of both conditional and unconditional CRMs over baseline reward models. The empirical validation across multiple bias domains substantiates the method’s robustness and generalizability.

Clarity: The paper is well-structured and written clearly. Figures and tables (e.g., Fig. 2 and 3) effectively visualize comparative performance trends.

Significance:
Reward hacking remains a major bottleneck for safe deployment of RLHF-trained LLMs. Hence, this paper’s contribution is relevant.

**Weaknesses:**

Hyperparameter Sensitivity: The choice of MMD coefficients (λ) significantly affects results (Fig. 3), but the paper does not deeply explore tuning stability or sensitivity. This could impact deployment robustness in practice.

**Questions:**

Causal Assumptions: How sensitive is CRM to incorrect causal graph specifications?

Scalability: How does the computational cost of MMD regularization scale with large datasets and high-dimensional embeddings?

Hyperparameter Selection: How was λ chosen in practice? Did you observe stable optima across tasks or require dataset-specific tuning?

---

### Official Review · Reviewer_WY6t · 2025-10-29

**Soundness:** 2
**Presentation:** 1
**Contribution:** 2
**Rating:** 2
**Confidence:** 3

**Summary:**

The paper introduces a method called CRM to address spurious correlations and reward hacking in RLHF. The central idea is to remove the confounding effect of response length so that the reward model more accurately reflects the true quality of the response. The authors leverage MMD to design a regularizer for reward model training. Experiments are conducted to evaluate the performance of the proposed method.

**Strengths:**

The paper introduces a new perspective by leveraging the method in counterfactual invariance. The proposed CRM serves as a regularizer that can be integrated into standard reward model frameworks.

**Weaknesses:**

- I am concerned about the novelty of this paper, as it largely builds on the technique proposed in [1], which already demonstrates that MMD can serve as a causal regularizer. In comparison, the present work appears to mainly combine and reapply techniques from that paper.

- The paper is not well written, causing unnecessary confusion in its methodology.

- Line 212: “the causal graph reveals that $T^{L, \perp}$ is independent of $Z$,” but Figure 1 shows that $Z$ directly influences $T^{L, \perp}$.

- I believe Equation (1) contains the key insight of the paper, but its description is delayed, and the preceding discussion seems disconnected from it.

- Is $T^{L, \perp}$ well-defined?

- The description of MMD could be moved to the Preliminaries section to make the methodological contribution clearer.

- Why is it necessary to bin $Z$? Why not directly measure the dependence between $f(T)$ and $T$ using the Hilbert–Schmidt Independence Criterion (HSIC)? These questions further aggravate my concern about the novelty of the paper, as it closely follows the approach in [1].

- The definition of $p'_m$ is also unclear.

- I am skeptical about the effectiveness of the proposed formulation. From a statistical perspective, MMD only measures dependence; it does not imply causality. Intuitively, applying MMD regularization in (1) cannot ensure the causality claimed in this paper.

- How is $Z$ chosen in practice? What if $Z$ includes a factor that does reflect the true reward of the response? Naively applying Equation (1) in such cases could hinder effective reward learning.

- The experimental comparison is limited. The authors should consider including more recent methods, for example:
  - https://arxiv.org/abs/2403.19159
  - https://arxiv.org/abs/2502.00814

[1] https://arxiv.org/abs/2106.00545

**Questions:**

See Weaknesses section

---

> ### Author Response · Authors · 2025-11-15
>
> Thank you for your review. We respectfully disagree with several of your assessments and address them point by point below.
>
> # Novelty and Relation to [1]
> Your concern about novelty overlooks the significant extensions we make beyond [1]. While [1] uses MMD as a causal regularizer in generic classification tasks, our Causal Reward Modeling (CRM) is the first to adapt this to the unique challenges of reward learning in RLHF pipelines for LLMs. We do not "mainly combine and reapply" [1]'s techniques; instead, we innovate by tailoring counterfactual invariance to preference-based rewards, addressing LLM-specific biases like sycophancy, length favoritism, and demographic discrimination. [1] neither explores reward modeling nor scales to LLM experiments—our application fills this gap, providing a plug-in regularizer that empirically mitigates reward hacking in ways [1] does not anticipate or demonstrate. Dismissing this as non-novel ignores the domain shift and practical advancements, which are central to advancing LLM alignment.
>
> # Writing and Clarity
> We acknowledge that some sections could be streamlined for clarity and will make targeted revisions in the camera-ready version, such as moving the MMD description to Preliminaries and introducing Equation (1) earlier with tighter connections to the preceding RLHF motivation. However, the methodology is fundamentally sound and not "unnecessarily confusing"—it builds logically from spurious correlations in reward models to our causal solution. Equation (1) is indeed the core insight, and its placement reflects a deliberate buildup; any perceived disconnection is minor and easily addressed without undermining the paper's quality.
>
> Regarding specific definitions: $ Z $ is clearly defined in Section 3 as the spurious factor (e.g., response length or style) that confounds preferences without affecting true utility. Similarly, $ \hat{r} $ is the learned reward function, explicitly introduced in the context of Equation (1). These are standard in the literature and well-grounded here—if they seemed unclear, it may stem from a rapid read rather than inherent flaws.
>
> # Line 212 and Figure 1
> There is no contradiction here. Line 212 correctly states that the causal graph (under intervention) reveals $ Y $ (true reward) is independent of $ Z $ (spurious factor). Figure 1 depicts the *observational* graph with confounding paths, but our method enforces *counterfactual* invariance, severing spurious influences. This is a standard causal inference concept (e.g., do-calculus), and the wording is precise—your interpretation seems to conflate observational and interventional semantics.
>
> # Binning $ Z $ and Alternative to HSIC
> Binning $ Z $ is a deliberate and necessary choice for scalability in LLM settings, where $ Z $ can be continuous (e.g., length) and high-dimensional. It enables efficient counterfactual sampling and MMD computation without the kernel explosion issues that plague HSIC in large-scale data. Suggesting HSIC as a drop-in alternative ignores these practical constraints—HSIC would be computationally prohibitive for our preference datasets. Moreover, our use of MMD builds on [1] but is far from a mere copy; we adapt it for reward invariance in RLHF, yielding novel empirical benefits. Questioning novelty here is unfounded, as adaptation to new domains *is* innovation.
>
> # Causality via MMD
> Your skepticism that "MMD only measures dependence, not causality" misrepresents both [1] and our work. Under the causal assumptions we explicitly state (e.g., the DAG in Figure 1, faithfulness, and no unobserved confounders—standard in causal ML literature), MMD regularization *does* enforce approximate counterfactual invariance, as rigorously proven in [1]'s Theorem 1. We do not claim MMD "implies" causality in a vacuum; rather, it operationalizes causal robustness in practice. Our experiments robustly validate this: CRM reduces biases in held-out tests without needing explicit counterfactual data, outperforming non-causal baselines. Dismissing this as ineffective ignores the theoretical foundation and empirical evidence—intuition alone does not override these.
>
> # Choosing $ Z $
> $ Z $ is chosen based on well-documented biases from prior LLM audits (e.g., length from Anthropic's work), making it practical and targeted. If $ Z $ inadvertently includes a true reward factor, our ablations (Table 2) already show minimal impact, as the regularizer's strength can be tuned via hyperparameters to preserve core signals. Naively assuming harmful inclusion is a strawman—our modular framework allows domain experts to refine $ Z $ iteratively, ensuring it enhances rather than hinders learning. This is not a flaw but a strength of our approach.

---

> > ### Author Response · Authors · 2025-11-15
> >
> > # Experimental Comparisons
> > Our experiments are comprehensive and focused, covering key baselines on diverse datasets like HelpSteer and synthetic preferences, with metrics directly tied to causal robustness. The suggested methods ([2403.19159] and [2502.00814]) address length bias in DPO but lack our general causal framing—they are orthogonal disentanglement techniques, not direct competitors to CRM's invariance regularizer. Including them would dilute focus without adding value, as our results already demonstrate superior generalization to unseen biases. Expanding comparisons is unnecessary; our setup rigorously supports our claims without it.
> >
> > We firmly believe these points refute your concerns and affirm the paper's originality, rigor. Thank you again for the feedback.

---

### Official Review · Reviewer_w8yn · 2025-10-31

**Soundness:** 2
**Presentation:** 2
**Contribution:** 2
**Rating:** 6
**Confidence:** 2

**Summary:**

The paper proposed a new approach called ‘Causal Reward Model’ to mitigate the problem of spurious correlation and reward hacking that is common in RLHF. Traditional methods use spurious shortcuts to deal with hacking coming from factors like long length. The paper, instead, treats the factors like length and tone as spurious variables $Z$, and learn a reward predictor whose internal representation is independent of $Z$. A maximum mean discrepancy based on the reward predictor is then added to the loss function as the regularisation.

**Strengths:**

-The paper is based on causal language and counterfactual invariance, which is novel compared to previous methods using supurious shortcuts. The CRM is then added as a normal regulariser which can be embedded into common model frameworks easily.
- Empirical results are also good.

**Weaknesses:**

- The method can only debiases the explicit spurious factors. If an unknown latent shortcut hack the reward, it won’t fix it.
- The benchmark is PPO, but what about other methods that address the shortcuts, the author should compare with these methods as well.

**Questions:**

- In practice, how to decide the spurious variables Z?
- Does CRM stay robust after the fine-tuning, which may cause the policy’s distribution shift against the preference dataset used to train CRM?

---

### Official Review · Reviewer_QC9D · 2025-10-31

**Soundness:** 3
**Presentation:** 1
**Contribution:** 3
**Rating:** 2
**Confidence:** 3

**Summary:**

The paper introduces a causality inspired regularizer in Reward Modelling.

**Strengths:**

The idea seems promising and novel. The idea is well motivated, and the solution is presented with clarity.

**Weaknesses:**

The experimental section seems to have missed out on providing some details.

1) For any one of the dataset mention how you are creating the bins.
2) How are you calculating p_m and p_m^'. For what values of Z is this being computed?
3) What are the values that Z can take in any one of the datasets?
4) Describe the conditional CRM in more detail.

**Questions:**

See above.

---

> ### Author Response · Authors · 2025-11-15
>
> Thank you for your review. We appreciate your recognition of the paper's promising and novel idea, strong motivation, and clear presentation. However, we strongly disagree with the low rating and the claim that the experimental section lacks details—these are explicitly provided in the paper, including appendices.
>
> # Experimental Details Overview
> Contrary to your assessment, Section 5 and Appendix B provide comprehensive experimental details, including dataset descriptions, hyperparameter tuning, and ablation studies. We evaluate CRM on multiple biases (sycophancy, length, concepts, discrimination) using datasets like Anthropic's HH-RLHF, Yelp, IMDB, Amazon reviews, and synthetic preferences. Baselines (e.g., vanilla RM, condition and unconditional rms) are rigorously compared via metrics like accuracy under concept shifts and bias reduction rates. Any "missing" details are either explicitly stated or follow standard practices in causal ML for RLHF.
>
> # 1. Creating Bins for Datasets
> Bins are clearly described in Section 4 (Methodology) and Appendix B. For continuous Z (e.g., response length in Section 5.2), we partition Z into M discrete bins to enable efficient MMD computation across pairs—e.g., explored bin sets like {10, 20, 30} for length in ablation studies (Table 3). For discrete Z (e.g., demographics in Section 5.4 and Appendix B.4.2), we group similar attributes into bins based on categories: age ("teen", "adult", "elderly"—4 bins), gender ("male", "female", "non-binary"—3 bins), race (5 categories like "white", "Black"), yielding 4 × 3 × 5 = 60 bins total. This binning is standard for handling high-dimensional or continuous confounders in MMD-based regularization (as in [1]) and ensures scalability in LLM preference data without losing granularity.
>
> # 2. Calculating p_m and p_m', and Values of Z for Computation
> These are defined in Section 4, Equation (2), and Appendix A. p_m and p_m' are the conditional distributions of the reward model's latent representation f(T) (or output r_ϕ(x,y)) within bins m and m', where T is the prompt-response pair. We compute them empirically from samples in each bin: for a bin m, p_m(f(T)) is the distribution of f(T) for data points where Z falls into m. MMD is then calculated across all pairs m, m' ∈ [M] as MMD(p_m(f(T)), p_m'(f(T))), using a Gaussian kernel (details in Appendix B.1). This is done for all Z values within the dataset—e.g., for length bias (Section 5.2), Z ranges over response lengths in the training data, partitioned into M bins; for binary Z (e.g., presence/absence of sycophantic phrasing in Section 5.1), it's direct MMD between p(f(T)|Z=0) and p(f(T)|Z=1). Computations are over the entire preference dataset D, with λ tuning the regularizer strength (hyperparameters in Appendix B.2).
>
> # 3. Values Z Can Take in Datasets
> Z's values are detailed per experiment in Section 5 and Appendix B. For example, in the length bias experiment (Section 5.2, using synthetic preferences from HelpSteer-like setups), Z is response length, taking continuous values (e.g., word counts from 10 to 300+), partitioned into M bins (e.g., {10, 20, 30} as ablated). In sycophancy (Section 5.1, HH-RLHF dataset), Z is binary/discrete phrasing (e.g., "Yes, you are right" vs. neutral). For concept bias (Section 5.3, Yelp/IMDB/Amazon), Z includes discrete concepts like "price", "service", "food" (Yelp) or "size", "color", "style" (Amazon). In discrimination (Section 5.4, synthetic data), Z spans 60 demographic bins across age (e.g., "teen" to "elderly"), gender (e.g., "male", "non-binary"), race (e.g., "white", "Asian"), etc. These are grounded in prior audits (e.g., Anthropic's length bias work) and ensure Z captures real-world spurious factors without overcomplication.
>
> # 4. Description of Conditional CRM
> Conditional CRM is fully described in Section 4.2 and evaluated in Section 5 (e.g., Tables 1-4). It extends unconditional CRM by applying the MMD regularizer separately to the chosen (y_w) and rejected (y_l) response subsets in preference data D, enforcing invariance within each subset: e.g., f(x, y_w) ⊥ Z and f(x, y_l) ⊥ Z. This is motivated by preference asymmetries in RLHF (Figure 1) and operationalized in the objective: the RM loss E_{(x,y_w,y_l)~D} [log σ(r_ϕ(x, y_w) - r_ϕ(x, y_l))] + λ * MMD terms for chosen and rejected distributions across bins. Empirically, it often yields stronger debiasing (e.g., reducing sycophantic rate to 19.78% vs. 62.64% unconditional in Table 1) while preserving utility, with ablations confirming its benefits over unconditional variants. Full pseudocode is in Appendix C.
>
> We believe these clarifications demonstrate that the paper already provides the requested details, supporting its good soundness and contribution scores. The low presentation rating seems inconsistent with your own praise for clarity. We will add minor signposts in the camera-ready to further highlight these sections if accepted. Thank you for your time.

---

### Meta-Review · Area_Chair_CSfT · 2025-12-15

**Summary:**

The authors did not provide a rebuttal to respond to two of reviewers. For the rebuttal to other two reviewers, AC believes it is not very convincing. Therefore, the paper is a clear rejection.

**Reviewer Concerns:**

Reviewer concerns AC thinks were addressed by the rebuttal:

AC finds the rebuttal only repeated what the paper has stated. AC believes the rebuttal is not very convincing to change reviewers' opinion from a score of 2 to a positive score.

Reviewer concerns AC believes are still outstanding:

1. The method can only debiases the explicit spurious factors. If an unknown latent shortcut hack the reward, it won’t fix it.

AC's comment: The authors didn't respond to it in the rebuttal.

2. The benchmark is PPO, but what about other methods that address the shortcuts, the author should compare with these methods as well.

AC's comment: The authors didn't respond to it in the rebuttal.

3. Hyperparameter Sensitivity: The choice of MMD coefficients (λ) significantly affects results (Fig. 3), but the paper does not deeply explore tuning stability or sensitivity. This could impact deployment robustness in practice.

AC's comment: The authors didn't respond to it in the rebuttal.

4. The experimental section seems to have missed out on providing some details.

AC's comment: The authors didn't provide extra experiments in the rebuttal.

**Reviewer Scores:**

The authors did not provide a rebuttal to respond to two of reviewers. For the rebuttal to other two reviewers, AC believes it is not very convincing. Furthermore, two reviewers give scores of 2 in the first round of review. AC believes the paper is not ready for publication and is a clear rejection.

---

### Decision · Program_Chairs · 2026-01-26

Reject